# Effect of Myofascial Therapy on Pain and Functionality of the Upper Extremities in Breast Cancer Survivors: A Systematic Review and Meta-Analysis

**DOI:** 10.3390/ijerph18094420

**Published:** 2021-04-21

**Authors:** Inmaculada Carmen Lara-Palomo, Adelaida María Castro-Sánchez, Marta María Córdoba-Peláez, Manuel Albornoz-Cabello, Lucía Ortiz-Comino

**Affiliations:** 1Department of Nursing, Physical Therapy and Medicine, University of Almeria, Road Sacramento s/n, 04120 Almeria, Spain; adelaid@ual.es (A.M.C.-S.); martamcordoba@gmail.com (M.M.C.-P.); 2Department of Physiotherapy, University of Sevilla, Avicena Street s/n, 41009 Sevilla, Spain; malbornoz@us.es; 3Department of Physical Therapy, University of Granada, Technological Park of Health Sciences, Avenue of Illustration 60, 18071 Granada, Spain; luciaoc@ugr.es

**Keywords:** breast cancer, myofascial release, functionality, pain, quality of life, range of motion

## Abstract

(1) Objective: The purpose was to analyze the effectiveness of myofascial therapy on musculoskeletal pain and functionality of the upper extremities in female breast cancer survivors, and to evaluate the changes in range of motion, quality of life, and mood state of these patients. (2) Methods: Systematic searches were performed on the MEDLINE/PubMed, Web of Science, Scopus, and Physiotherapy Evidence Databases for articles published until October 2020, in order to identify randomized controlled trials which analyzed the effectiveness of myofascial therapy as compared to a control group, passive treatment, placebo, or another intervention, and allowed co-interventions on female breast cancer survivors. Two reviewers examined the sources individually, calculated the risk of bias and extracted the data (PROSPERO number CRD42020215823). (3) Results: A total of eight RCTs were included. The results suggested that myofascial therapy does not have a greater statistically significant immediate effect on pain intensity (SMD: −0.15; 95% CI −0.48, 0.19), functionality (SMD: −0.17; 95% CI −0.43, 0.09) and range of motion in flexion (SMD: 0.30; 95% CI −0.13, 0.74) than an inactive, passive treatment or another intervention. However, a statistically significant result was observed for the abduction shoulder in favor of the experimental group (SMD: 0.46; 95% CI 0.05, 0.87; *p* = 0.03). (4) Conclusion: In general, although we found greater overall effects in support of the intervention with myofascial therapy than other control groups/types of interventions, the subgroup analysis revealed inconsistent results supporting myofascial therapy applied to breast cancer survivors.

## 1. Introduction

Of the 18.1 million cases of cancer globally diagnosed [1], breast cancer accounts for 11.6%, being the most common cancer among women [2,3]. In Spain, this incidence increases to 28.7%, and one in eight women is at risk of developing breast cancer [4]. The rising incidence of breast cancer in developed countries is mainly due to demographic factors, lifestyle, and reproduction rates [5]. Despite this high incidence, thanks to developments in early detection techniques, as well as rapid implementation of treatment protocols [6], its survival rate is over 90% [7].

Treatment of breast cancer generally involves a combination of different methods, and may produce toxicities, which can be cumulative and difficult to separate clinically, including surgery, radiation, chemotherapy, hormonal therapy, and/or targeted therapy [8,9]. Surgical treatment for breast cancer includes breast-conserving surgery, combined with radiation, or mastectomy with or without radiation and with or without immediate/deferred reconstruction [9]. Breast-conserving therapy, consisting of breast-conserving surgery (lumpectomy) plus radiation, is the standard treatment for most women with stage I and II breast carcinomas [10]. However, all survivors are at risk of suffering from side effects of treatment in the short- or long-term, such as hemorrhage, infection at the surgical site, weakness in the arm or shoulder, restricted movement, swelling, numbness, pain, and lymphedema [11,12,13,14,15].

Radiation therapy is an adjuvant therapy that is used in 50% of patients, which may lead to fibrosis in the adjacent tissue [8]. Radiation-induced fibrosis is a form of damage to normal tissues after radiation therapy. It can affect the underlying fascia, muscles, organs, and bones on both the affected and unaffected sides, cause persistent symptoms and aesthetic disfigurement, thereby affecting quality of life [16,17,18,19].

Furthermore, the dissection of the axillary lymph nodes, the radiation on the regional lymph nodes and the patient’s preoperative body mass index, among others, are considered to be factors that contribute to the development of one of the most frequent side effects of breast cancer: lymphedema [20]. The risk of getting lymphedema after overcoming cancer is between 6% and 45% [21,22,23,24] and, of these cases, 90% arise between 18 and 24 months after treatment [20,25]. Additionally, lymphedema is associated with other symptoms, such as pain, bloating, pressure, fatigue, limited joint movement, mainly in the abduction of the shoulder and elbow bending, and the subsequent reduction in use of the affected limb [22,26]. Conversely, postoperative pain is another side effect which occurs in at least half of women who have undergone surgery between 6–15 months after the operation [27,28]. The prevalence of neuropathic pain is 24% nine months after surgery [28].

While improvement in diagnostic processes and in the choices available for medical treatment to reduce possible long-term effects have led to a higher survival rate after breast cancer diagnosis, new challenges have arisen in addressing these effects in healthcare systems [29]. There are several studies which suggest the use of physical therapy to treat side effects of breast cancer following medical treatment. Among the most highly-recommended therapies are mobilization, active exercises or active-assisted exercises, and manual therapy [30,31,32]. Myofascial release is found under the scope of manual therapy, and is a low-impact, long-term treatment with the aim of restoring the length of the fascia, eliminating functional limitations and reducing the perception of pain to improve the function of the locomotor system [33]. Numerous clinical trials have demonstrated the benefits of myofascial therapy on different populations, showing an improvement in range of motion and a decrease in perceived pain [34,35,36,37]. However, to date, no systematic reviews nor meta-analyses have been carried out about the effects of myofascial therapy on the treatment of the side effects derived from the medical treatment of breast cancer, whereas because of its characteristics as a manual therapy specialty as well as the absence of secondary side-effects after its use [33], it seems an adequate technique to manage with breast cancer survivors’ impairments.

Thus, the objective of this study is to perform a systematic review, together with a meta-analysis, to check the effects of myofascial therapy on female breast cancer survivors’ pain, functionality, joint range of motion, and mood state. We hypothesize that myofascial release is an adequate approach to improve these factors.

## 2. Materials and Methods

### 2.1. Protocol and Registry

A systematic review and meta-analysis was carried out, taking into account the items on the recent declaration of ‘preferred reporting items for systematic reviews and meta-analyses’ (PRISMA) [38]. Systematic review registration: http://www.crd.york.ac.uk/PROSPERO (accessed on 21 April 2021). PROSPERO registration number: CRD42020215823 (23 November 2020).

### 2.2. Search and Information Sources

The bibliographic search was performed during the months of March through October 2020 on the Medline (through the platform PubMed), Scopus, Web of Science, and PEDro databases. The search strategy was based on terms registered on the MeSH list (“breast”, “cancer”, and “myofascial”) combined with the Boolean operator (AND), adapted to the characteristics of each database. Any duplicates that were identified in the multiple database searches were removed. Additionally, the reference lists for the included studies were also examined and experts in the field were contacted (for example, authors of the included studies) to obtain additional information or information which was not implicit in the published trials.

### 2.3. Inclusion and Exclusion Criteria

The inclusion and exclusion criteria were defined using the PICO process [39] (Patient, Problem or Population, Intervention, Comparison, Control or Comparator, Outcome(s)).

#### 2.3.1. Types of Studies

Randomized controlled clinical trials (RCTs) were included. Quasi-experimental controlled trials were excluded. In addition, the following inclusion criteria were taken into account: RCTs published in the last 20 years (from 2000 onwards), written in English, available in full-text version, focused on the ongoing effects of breast cancer intervention with myofascial techniques. All studies that did not meet these characteristics were excluded.

#### 2.3.2. Types of Participants

Studies with female subjects who had completed breast cancer treatment at least two months prior and had upper limb or neck pain. An exception was made with one of the studies, which was related to palliative care, but was included, as the same aspects were assessed as in the other studies. The age of participants and tumor location were not considered as criteria.

#### 2.3.3. Types of Interventions

Studies were included in which side effects derived from breast cancer treatment were treated with myofascial therapy. Studies that treated these effects with myofascial therapy accompanied by other manual or physical therapies were not excluded, nor were studies restricted by the duration of treatment, frequency or type of techniques applied in the treatment. Studies comparing myofascial therapy with any other intervention or no intervention were included. We also included any study which compared myofascial therapy accompanied by other therapies, with only the application of those other therapies. All habitual medication was allowed to be taken during the studies.

#### 2.3.4. Types of Outcome Measures

Studies that evaluated one or both of the following aspects as primary measures were chosen: pain and functionality of the shoulder affected. For pain, the studies should have used the Visual Analogue Scale (VAS), or a comparable numerical scale, and in the evaluation of functionality, the Disabilities of the Arm, Shoulder and Hand (DASH), or scale should have been used. As a secondary outcome measure, the following were also taken into consideration: the evaluation of shoulder mobility using goniometry; the mood state of the participants evaluated, for example, with the Profile of Mood States (POMS) scale; and quality of life measured using the Health Questionnaire SF-36. The results were collected in three specific time periods: immediately following treatment, short-term (≤3 months), medium-term (between six and nine months), and long-term (≥12 months).

### 2.4. Study Selection

Two reviewers (I.C.L.-P., M.M.C.-P.) used the pre-specified criteria to select relevant titles, abstracts and full articles. An article was deleted if it was determined that it did not meet the inclusion criteria. If there were any hesitations about the selection decisions, a third reviewer (A.M.C.-S.) was consulted. Once the review was completed, the search strategy was repeated, in case any new studies had been published, and they were analyzed to assess their inclusion. The last search was carried out in November 2020.

### 2.5. Data Extraction and Management

Firstly, the titles and abstracts of the references retrieved from the searches were selected. The full text was obtained for references which the authors considered potentially relevant. Full-text references were then independently evaluated for inclusion according to the inclusion criteria for considering studies for this review.

To manage the data, a data summary sheet was created, based on the Cochrane recommendations. The data extracted were: author and year, type of study, number and type of patients, type of intervention, number and duration of treatment sessions, outcome measures, and primary outcomes found among groups. If key information was missing from the study report, the authors of the report were contacted to obtain this information.

### 2.6. Risk of Bias Assessment

We used the recommendations from the Cochrane Collaboration to evaluate the risk of bias for all of the articles. Each item was evaluated with the objective of discerning whether the trials eligible for inclusion in this review were valid enough for their results to be interpreted. The items evaluated were: specification of selection criteria, random sequence generation, homogeneity among groups, allocation concealment, blinding of participants and personnel, blinding of the outcome assessors, incomplete outcome data, selective reports, and other biases.

Two reviewers (I.C.L.-P., L.O.-C.) independently assessed the risk of bias for each of the articles selected for the current study. An arbitrator (A.M.C.-S.) was consulted to settle any disagreements.

Each item was classified as “high risk”, “low risk”, or “unclear risk” of bias. A sensitivity analysis was performed on the primary results to explore the effects of including and excluding trials with a high risk of bias (sensitivity analysis).

### 2.7. Statistical Analysis

A heterogeneity analysis of the selected studies was performed. In terms of continuous data and dichotomous data, effect sizes were measured using standardized mean difference (SMD) and 95% confidence interval (CI), or risk ratio (RR) with 95% CI, respectively. Heterogeneity within RCTs was examined using the I^2^ test, considering I^2^ ≥ 50% as a sign of substantial heterogeneity. Once there were >2 homogeneous studies, RevMan 5.4 (Cochran Collaboration, London, UK) software was used to perform meta-analyses. Sensitivity analyses were conducted for the robustness of the result of meta-analyses.

## 3. Results

### 3.1. Selection of Studies

The initial search retrieved 181 potential articles, of which 112 were excluded after reviewing their titles and abstracts, and 27 were duplicates, leaving 42 full-text articles to be reviewed. After the application of the inclusion and exclusion criteria, 34 of these articles were eliminated, leaving eight studies for this systematic review and meta-analysis. Figure 1 illustrates the different phases of the review, using the eligibility and data-synthesis PRISMA flow diagram.

### 3.2. Characteristics of Studies Included

Eight RCTs were included, with a total of 333 participants [40,41,42,43,44,45,46,47]. The study developed by Fernández-Lao et al., divided into two publications [40,41] contained the smallest sample-size, with 20 participants; whereas the one with the largest sample size was the study by Groef et al. [42] with 147 participants.

Regarding the location where the studies took place, three were performed in Spain [40,41,43,44], three in Belgium [42,45,46] and another in the Midwestern United States [46]. All of the studies reported the place of recruitment and methods used. The study developed by Fernández-Lao et al. were was performed on patients from the Oncological Unit at the Virgen de las Nieves Hospital in Granada, Spain [40,41]; one at the University of Granada [43]; one in Valencia, Spain, at the Spanish Association Against Cancer (AECC) [44]; three at the University Hospital in Leuven, Belgium [42,45,46]; and in the study carried out in the United States, the women interested in taking part in the study contacted the coordinator to participate [47]. Table 1 shows the main characteristics of each of the studies.

### 3.3. Characteristics of the Participants

All of the studies contained participants diagnosed with breast cancer. No distinction or mention of race is made in any study. The age of the patients was detailed in all of the studies, ranging from 21–65 years old [40,41,42,43,44,45,46,47]. In addition, the stage of patients’ cancer was considered as inclusion criteria by Fernández-Lao et al., and was their study was carried out on subjects with cancer in stages I-IIIA [40,41]. Only the study by Serra-Añó et al. [44] considered the type of surgery the patients had undergone (only conservative/partial) when choosing study participants.

### 3.4. Characteristics of the Interventions

All of the studies had two comparison branches. The studies compared myofascial therapy with: educational sessions on healthy lifestyles, focusing on nutrition, relaxation techniques, and exercise, with advice on how to improve quality of life after cancer [40,41], with a standard physical therapy program and consistent placebo intervention with bilateral static hand placements on the upper body and arm [42,45,46], with pulsed shortwave therapy [43], and with relaxing Swedish massage which avoided the affected area [47].

The types of myofascial interventions used, although all of them were manual interventions, varied among the studies. A myofascial induction protocol centered on the neck and shoulder following the Barnes approach [40,41], trigger point treatments on upper limbs and adhesions in the pectoral muscle, cervical region, diaphragm, and scars [42,45,46], myofascial massage specific to the chest, thorax, and shoulder of the affected side [46], and myofascial release [43,44] were all used.

With regard to the number and length of sessions and the duration of the therapy, the studies were very heterogeneous. Fernández-Lao et al. conducted two 40-min sessions separated by an interval of two and three weeks [40,41], two studies conducted two 30-min sessions per week for eight weeks [42,47] two conducted two 30-min sessions per week for 12 weeks, later reduced to one session per week [45,46], one a 50-min session per week for four weeks [44], and finally, one study conducted two 30-min sessions four weeks apart [43] (see Table 2).

### 3.5. Outcome Measures

#### 3.5.1. Primary Measures

Pain: All but one of the studies evaluated pain or pain-related outcomes [41]. In the majority of cases, it was measured using a visual analog scale (VAS) [42,43,44,45,47]; in two of the studies, quantitative and qualitative aspects of pain were assessed using the McGill Pain Questionnaire [42,45]. Furthermore, three studies measured pain threshold with pressure from an algometer on different muscle points [40,42,46]. Pain was assessed using a scale from 0 to 30 [47], a scale from 0 to 10 [43,44], and a scale from 0 to 100 [42,46]. One study measured pain in the cervical spine, the affected limb and non-affected limb [43], and two measured pain in the shoulder, neck region, arm, armpit, trunk side and breast region [42,46]. The unit of measurement differed between the studies, as one used kPa [40], and the other two used kg/cm^2^ [42,46].

Shoulder functionality: the majority of the studies measured functional status using the Disabilities of the Arm, Shoulder and Hand (DASH) scale [41,43,44,45]; for all of them, total scores were 0–100, with a higher score indicating a greater disability. One study measured shoulder functionality on an unvalidated scale ranging from 0 to 40, where 0 was no difficulty and 40 was severe difficulty [46].

#### 3.5.2. Secondary Measures

Shoulder mobility: Three studies evaluated shoulder mobility using manual goniometry in degrees [43,44,45]. One study measured flexion, abduction and active external and internal rotation on the affected and non-affected side [43]. Another also took into account extension and adduction but only of the affected side [44], and another measured the flexion and abduction, and the upward scapular rotation [45].

Mood state: Two studies evaluated the mood states of participants with the Profile of Mood States (POMS) scale [41,43]. Another study evaluated depression with the Patient-9 Health Questionnaire (PHQ-9), and one study linked participants’ saliva flow rate and the amount of immunoglobulin A and cortisol to stress levels and the patients’ attitude towards the intervention.

Quality of life: Five of the eight chosen studies evaluated participants’ quality of life [42,44,45,46,47], although the scales that were used varied among the studies. The most commonly-used scale was the SF-36 [42,45,46], although one study used the short version, SF-12 [47], and another one used the Functional Assessment of Cancer Therapy-Breast (FACT-B + 4) scale.

### 3.6. Follow-Up

More than half the studies limited their follow-up to immediately following the intervention [40,41,43,44,47]. Just two studies followed-up between ≥6 months and ≤9 months [45,46] and three studies also followed-up long-term (>9 months) [42,45,46].

### 3.7. Risk of Bias in the Studies Included

The results of the risk of bias (RoB) analysis for the individual studies are summarized in Figure 2 and Figure 3. In total, approximately one-third of the studies were considered to have a moderate RoB.

### 3.8. Effects of the Interventions

#### 3.8.1. Myofascial Therapy vs. Placebo Treatment or Other Intervention at Post-Immediate

Figure 4, Figure 5, Figure 6 and Figure 7 show the estimated primary effect size (4–8 weeks post-treatment) of the intervention with myofascial therapy alone or combined, compared to an inactive control, placebo treatment or other intervention with physical therapy for the outcomes of pain intensity, functionality, and range of motion in flexion and abduction.

Pain Intensity

This sub-analysis included four trials [42,43,44,45] with 242 participants. No significant differences were observed between the effects of myofascial therapy alone or in combination with a standard physical therapy program and placebo intervention or a standard physical therapy program (Figure 4. SMD −0.15, 95% CI −0.48 to 0.19, I^2^ = 28%).

2.Functionality

Meta-analyses of three studies [42,44,46] with 121 participants revealed that there were no statistically significant differences between the two groups on post-immediate functionality (Figure 5, SMD −0.17, 95% CI −0.43 to 0.09, I^2^ = 0%). However, the results were in favor of the group treated with myofascial therapy.

3.Range of motion

This subgroup involved 3 trials [43,44,45] covering 95 participants. Meta-analysis result demonstrated that myofascial therapy does not have a greater statistically significant im-mediate effect on range of motion in flexion (Figure 6, SMD 0.30, 95% CI −0.13 to 0.74, I^2^ = 11%) than a placebo treatment or other intervention. However, a statistically significant result was observed for shoulder range of motion in abduction in favor of the experimental group (Figure 7, SMD 0.46, 95% CI 0.05 to 0.87, I^2^ = 0%; *p* = 0.03).

Low-moderate quality evidence indicates that myofascial therapy does not have a greater statistically significant immediate effect on pain intensity, functionality, and range of motion in flexion than a placebo treatment or other intervention at post-immediate follow up. The estimations of the primary effect show small or null statistically significant heterogeneity between the effect size of the studies included in this analysis; the statistic I^2^ ranged from 0% “low threshold” and 28% “low-moderate threshold”.

#### 3.8.2. Myofascial Therapy vs. Placebo Treatment or Minimal Intervention at Post-Immediate

Figure 8, Figure 9 and Figure 10 show the results of the estimated primary and secondary effects as forest-plots grouped separately by outcome measure type, immediately following the intervention with myofascial techniques.

Pain intensity

Meta-analyses of two studies [43,44] with 45 participants, demonstrated that the effect of myofascial therapy was no superior to placebo treatment nor minimal intervention for alleviating post-immediate pain intensity (Figure 8, SMD −0.27, 95% CI −0.86 to 0.32, I^2^ = 0%).

2.Range of motion

Meta-analyses of two studies [43,44] showed that range of motion in flexion, abduction, internal and external rotation (Figure 9, SMD 0.23, 95% CI −0.07 to 0.52, I^2^ = 0%) was similar between the two groups.

3.Mood state

This sub-analysis of two trials [41,43] with 41 participants did not obtained statistically significant differences between groups (Figure 10, SMD −0.12, 95% CI −0.35 to 0.11, I^2^ = 0%).

Low-quality evidence indicates that applying myofascial techniques does not have a greater immediate effect on pain intensity, range of motion in flexion, abduction, internal and external rotation, and mood state, than a placebo or a minimal intervention. For the three outcome measures, the I^2^ statistic was 0%, null statistic heterogeneity.

#### 3.8.3. Myofascial Technique (Physical Therapy and Myofascial Therapy) vs. Other Interventions (Physical Therapy and Placebo) at Post-Immediate, Medium Term and Long Term

Figure 11, Figure 12, Figure 13, Figure 14, Figure 15, Figure 16, Figure 17, Figure 18 and Figure 19 show the results as forest-plots grouped by primary and secondary effects, and separated by the duration of the effects after completing the myofascial intervention.

Pain Intensity

This meta-analysis included two studies and 197 participants [42,46]. The point estimates for the SMD between the groups varied from 0.11 to 0.20 (Figure 11, Figure 12 and Figure 13. 95% IC; I^2^ = 0% to 71%) in the short-, medium-, and long-term; in all cases. The results of this meta-analysis indicate that there is not a statistically significant effect for pain intensity in favor of myofascial therapy over other types of interventions.

2.Functionality

Two studies with 197 participants were meta-analyzed for shoulder functionality [42,46]. The estimates varied from 0.15 to 0.19 (SMD) (Figure 14, Figure 15 and Figure 16. 95% IC; I^2^ = 0%) and although all of them favored the myofascial therapy, their differences were not statistically significant.

3.Quality of life

The result of the meta-analysis of two studies [42,46] demonstrated there were no statistically significant differences between the two groups for quality of life at any time (Figure 17, Figure 18 and Figure 19), grouped estimates ranged from 0.03 to 0.15 (SMD) (95% IC; I^2^ = 22% to 73%).

Moderate-high quality evidence indicates that myofascial therapy has no greater immediate, medium- and long-term effect on pain intensity, functionality, and quality of life than other forms of intervention.

The grouped estimates show little statistical heterogeneity among the studies’ effect sizes. Of the six comparisons of pain and disability, five showed a null I^2^ statistic, only immediate post-treatment pain was above the ‘high’ threshold of 75%.

## 4. Discussion

### 4.1. Summary of Evidence

We set out to conduct a unique, up-to-date review on the impact of myofascial therapy on breast cancer survivors. We found 8 published RCTs that evaluated the effects of myofascial therapy on these patients, although two studies could not be meta-analyzed. The study by Massingill et al. [47], could not be analyzed in the meta-analysis because of incomplete data in the results, and that of Groef et al. [45], presented duplicate results that were published in another of the studies included. Ultimately, for the qualitative analysis, six publications were considered, including five studies (as those publications by Fernández-Lao et al. [40,41] shared the same participants). Thus, a reduced number of patients than previously mentioned, 262 participants total, were included.

In general, we found that interventions with myofascial therapy as the sole intervention or combined with physical therapy, do not generate, with low or moderate-quality proof, a significant improvement on pain intensity, functionality, range of motion and mood state in female survivors of breast cancer compared to an inactive control group or placebo or in comparison with other interventions.

There were greater general effects in favor of myofascial therapy than other controls/interventions, but the subgroup analyses revealed inconsistent, insignificant results. Only the range of motion in abduction showed statistical significance in favor of the experimental group. Here, myofascial therapy is at least equivalent to other forms of intervention.

Several factors should be considered when interpreting our findings for clinical recommendation and implementation. Although there was no statistical significance, all of the studies individually reported a clinically relevant improvement in favor of the experimental group with myofascial intervention. The effectiveness of myofascial therapy is unclear because the subgroup analyses were hampered by the small number of studies that included the same outcome measures and follow-up beyond immediately following the intervention. Very few studies reported comparable estimates regarding secondary outcome measures to calculate the effect of using myofascial therapy.

### 4.2. Agreements or Disagreements with Other Studies or Reviews

Myofascial therapy and its effect on certain populations has been briefly described in literature. Other meta-analyses that have attempted to demonstrate the efficacy of myofascial therapy have also faced the challenge of only finding articles of low methodological quality, and as a result, a small number of articles are included in the meta-analysis. In 2016, Webb and colleagues [48] did not find conclusive results after their meta-analysis about the efficacy of myofascial therapy on joint range of motion and perceived pain. A recently-published article also obtained outcomes that tend towards the statistical significance of myofascial techniques for the improvement of joint range of motion, without these generating a clinically relevant change [49]. Our article correlates with this article, as obtaining statistical significance in the improvement of the range of motion in shoulder abduction reaffirms myofascial therapy as an effective technique in the improvement of this specific variable. Additionally, regarding survivors of breast cancer, other systematic reviews have been carried out with meta-analyses using the generic term “manual therapy”, which encompasses myofascial therapy. In the review carried out by Pinheiro and colleagues (2019), manual therapy does prove to be effective to decrease musculoskeletal pain [50]. However, this author could only include five articles in their meta-analysis, again, due to the limiting characteristics of the trials carried on manual therapy and cancer. These results are contradictory to those found in the review by Groef et al. (2015), that when analyzing the effectiveness of various postoperative physical therapy modalities, including myofascial therapy for the treatment of pain and range of motion of the shoulder in breast cancer, reported that to date, no RCT had reported on the effectiveness of myofascial therapy started in the postoperative phase after breast cancer treatment [30]. Despite the differences found regarding pain, our findings showed no improvement using myofascial therapy as opposed to other therapies on quality of life, which is consistent with the findings obtained in the review by Pinheiro and colleagues (2019) [50].

Nevertheless, despite not obtaining clear, insightful results in recent years on the clinical evidence of manual therapy and, thus, myofascial therapy, these techniques continue to be prescribed as treatment for pain management in cancer patients [51], and also for addressing other symptoms such as anxiety or altered mood states, due to their non-invasive nature and the absence of negative side effects. Myofascial therapy, described by Pilat in 2003 [52], is a relatively recent technique, which, despite its frequent use in clinical physical therapy, has not been explored deeply or frequently enough to test its effects on different populations. Thereby, as shown in this review, there are many different approaches of myofascial therapy. In order to understand whether myofascial therapy is a useful rehabilitation approach for this population, these differences should also be objective of future randomized controlled trials, to check the superiority of specific methods among others.

## 5. Study Limitations

Our review was limited by the small number published trials to date, the impossibility of blinding participants and physical therapists, the small sample size, and non-homogeneous follow-up of the included studies.

The outcomes for pain intensity, functionality, and range of motion should be considered key to evaluating the effects of myofascial therapy on the study population, as these factors are the main causes of the emotional burden of the disease [53]. Many of the included studies did not report these results and, when reported, they were measured at different time periods. The lack of standardization measurements makes quantitative synthesis of the body of evidence problematic.

The inconsistent nature of data collection and reporting made it difficult to draw conclusions regarding medium- and long-term results, as half of the studies only collected data immediately following the intervention.

For example, in the comparison between myofascial therapy versus a placebo or minimal intervention, the studies reported on the outcome measures of pain, range of motion and mood state only immediately following treatment. The fact that this data was not reported in a comparable manner over time limits our ability to estimate the true effect of myofascial therapy on a critical outcome.

## 6. Conclusions

While our subgroup analyses show non-significance between groups, the results are inconclusive. The choice of myofascial therapy over other control groups with/without intervention for breast cancer survivors is likely to result in a positive effect on pain intensity, functionality, and range of motion. Myofascial therapy is also likely to have a beneficial effect on quality of life and mood state outcomes. However, given the results obtained, it would seem that there is little to gain from referring these patients to myofascial treatment. Despite this, the volume of evidence is small and additional similar studies are likely to greatly change the estimation of the effectiveness of myofascial therapy versus inactive controls, placebo, or other physical therapy interventions. Future studies are needed to confirm whether myofascial therapy is useful or not to man-age breast cancer survivors’ sequelae.

## Figures and Tables

**Figure 1 ijerph-18-04420-f001:**
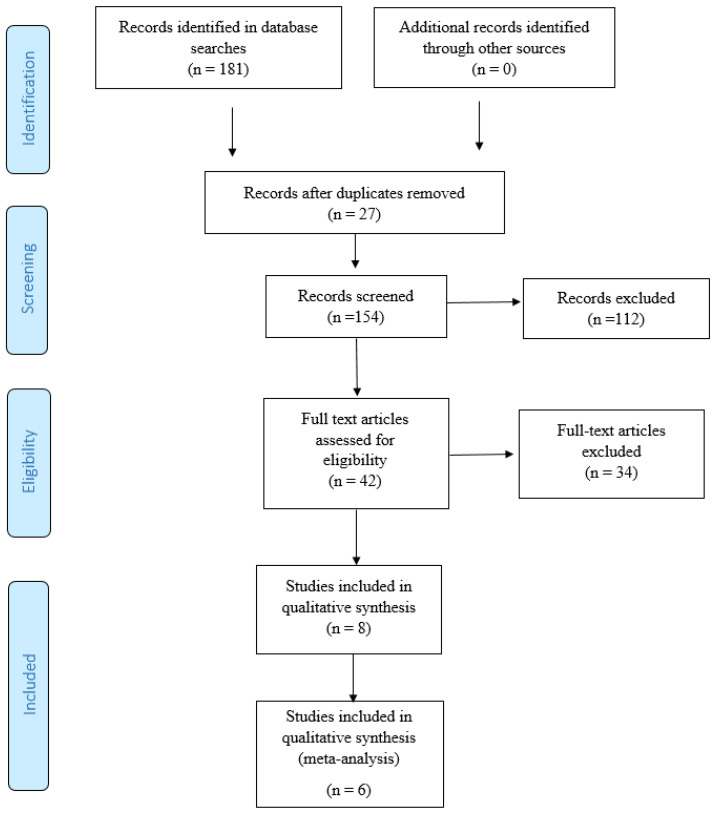
Eligibility and data synthesis: PRISMA flow diagram.

**Figure 2 ijerph-18-04420-f002:**
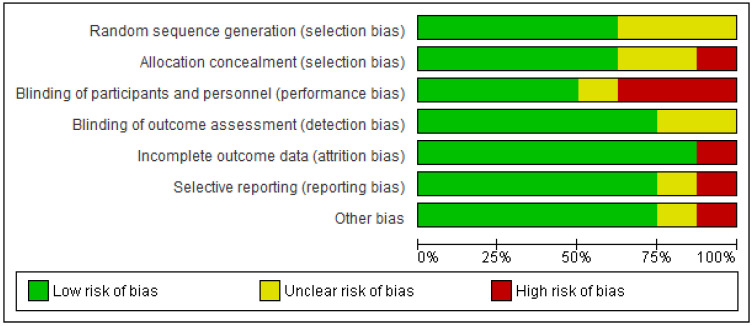
Risk of bias graph: review authors’ judgements about each risk of bias item presented as percentages across all included studies.

**Figure 3 ijerph-18-04420-f003:**
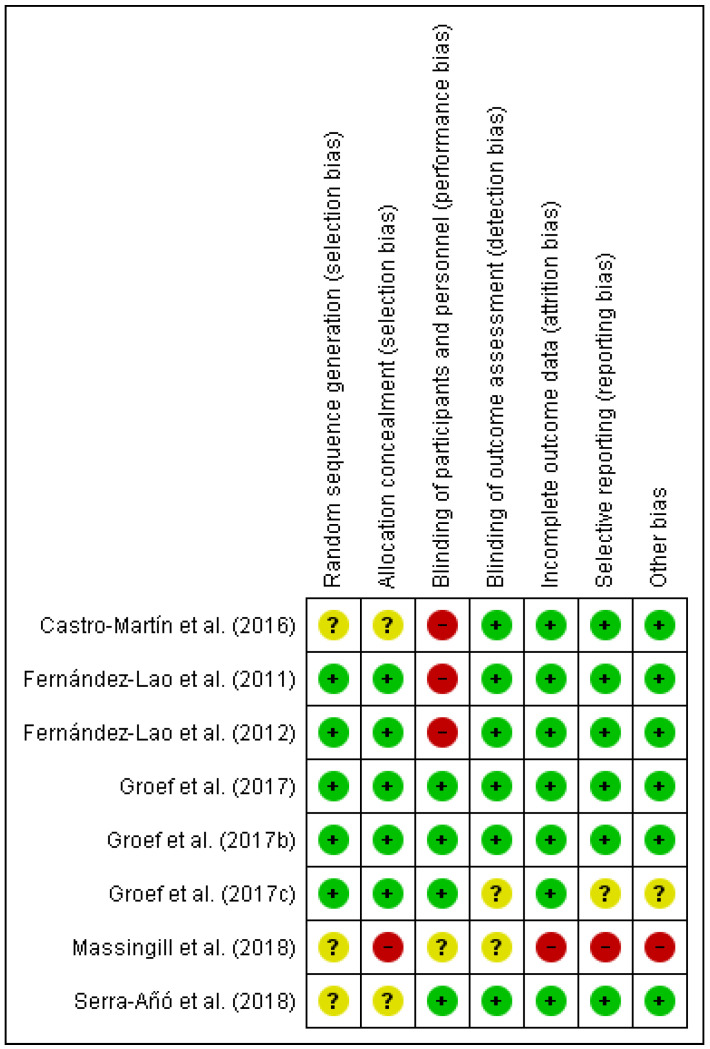
Risk of bias summary: review authors’ judgements about each risk of bias item for each included study.

**Figure 4 ijerph-18-04420-f004:**
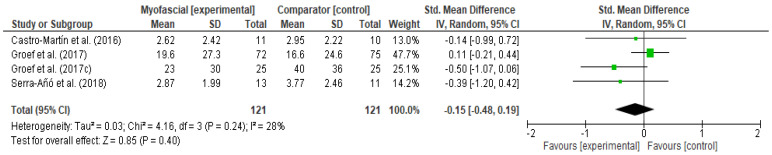
Forest plot of comparison: 1 Myofascial therapy vs. placebo treatment or other intervention, outcome: 1.1 Pain Intensity-Post immediate.

**Figure 5 ijerph-18-04420-f005:**
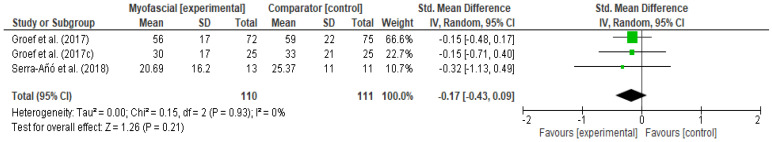
Forest plot of comparison: 1 Myofascial therapy vs. placebo treatment or other intervention, outcome: 1.2 Functionality-Post-immediate.

**Figure 6 ijerph-18-04420-f006:**
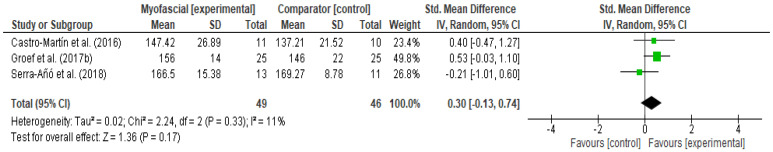
Forest plot of comparison: 1 Myofascial therapy vs. placebo treatment or other intervention, outcome: 1.3 Range of motion-Flexion-Post immediate.

**Figure 7 ijerph-18-04420-f007:**
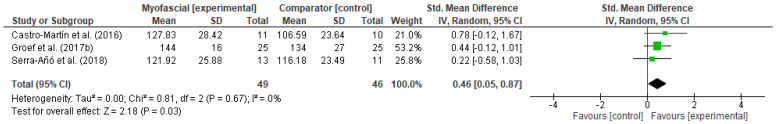
Forest plot of comparison: 1 Myofascial therapy vs. placebo treatment or other intervention, outcome: 1.4 Range of motion-Abduction-Post-immediate.

**Figure 8 ijerph-18-04420-f008:**
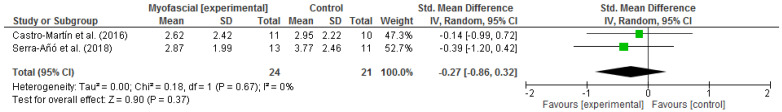
Forest plot of comparison: 2 Myofascial therapy vs. placebo treatment or minimal intervention, outcome: 2.1 Pain Intensity-Post immediate.

**Figure 9 ijerph-18-04420-f009:**
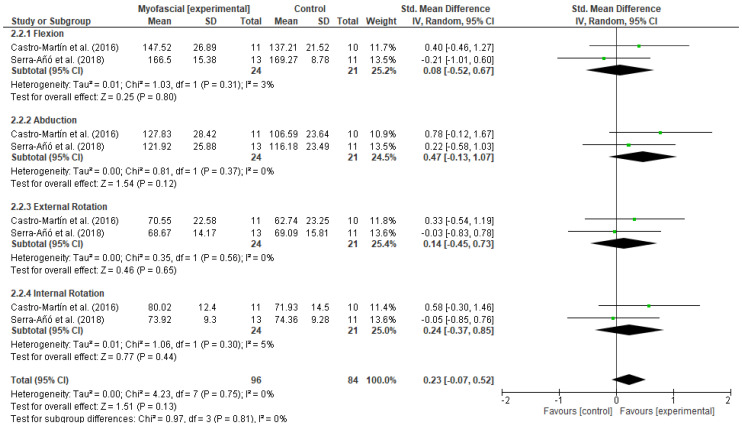
Forest plot of comparison: 2 Myofascial therapy vs. placebo treatment or minimal intervention, outcome: 2.2 Range of motion-Post-immediate.

**Figure 10 ijerph-18-04420-f010:**
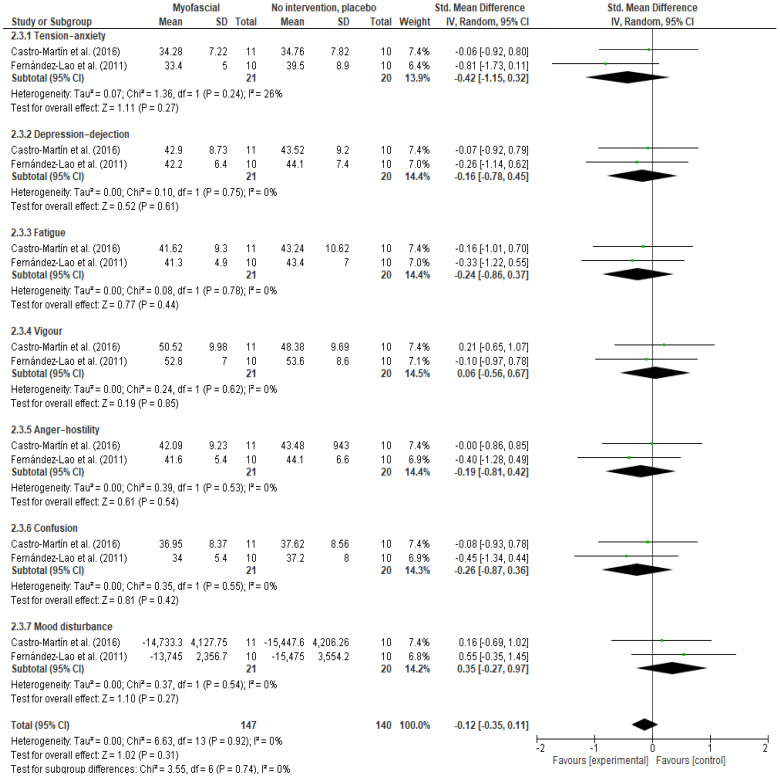
Forest plot of comparison: 2 Myofascial therapy vs. placebo treatment or minimal intervention, outcome: 2.3 Mood States-Post-immediate.

**Figure 11 ijerph-18-04420-f011:**
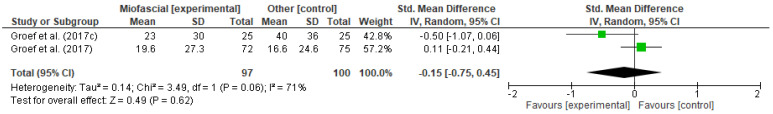
Forest plot of comparison: 3 Myofascial therapy (physical therapy and myofascial therapy) vs. Other intervention (physical therapy and placebo), outcome: 3.1 Pain Intensity-Post immediate.

**Figure 12 ijerph-18-04420-f012:**
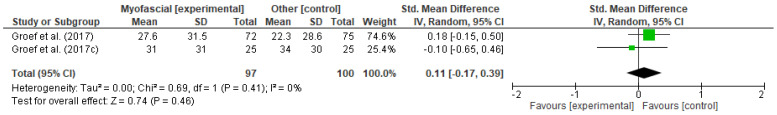
Forest plot of comparison: 3 Myofascial therapy (physical therapy and myofascial therapy) vs. Other intervention (physical therapy and placebo), outcome: 3.2 Pain Intensity-Medium term.

**Figure 13 ijerph-18-04420-f013:**
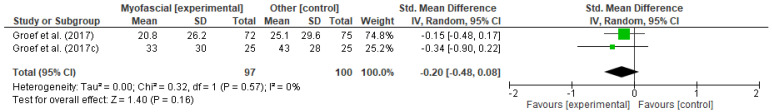
Forest plot of comparison: 3 Myofascial therapy (physical therapy and myofascial therapy) vs. Other intervention (physical therapy and placebo), outcome: 3.3 Pain Intensity-Long term.

**Figure 14 ijerph-18-04420-f014:**
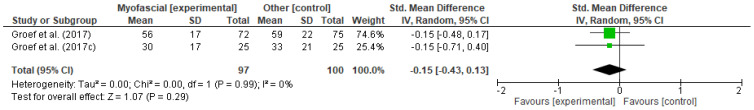
Forest plot of comparison: 3 Myofascial Therapy (physical therapy and myofascial therapy) vs. Other intervention (physical therapy and placebo), outcome: 3.4 Functionality-Post immediate.

**Figure 15 ijerph-18-04420-f015:**
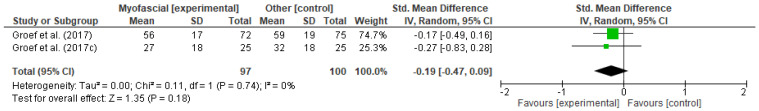
Forest plot of comparison: 3 Myofascial therapy (physical therapy and myofascial therapy) vs. Other intervention (physical therapy and placebo), outcome: 3.5 Functionality-Medium term.

**Figure 16 ijerph-18-04420-f016:**
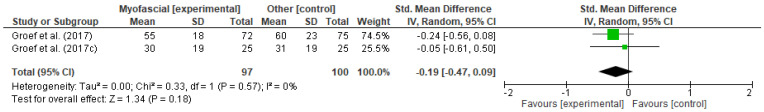
Forest plot of comparison: 3 Myofascial therapy (physical therapy and myofascial therapy) vs. Other intervention (physical therapy and placebo), outcome: 3.6 Functionality-Long term.

**Figure 17 ijerph-18-04420-f017:**
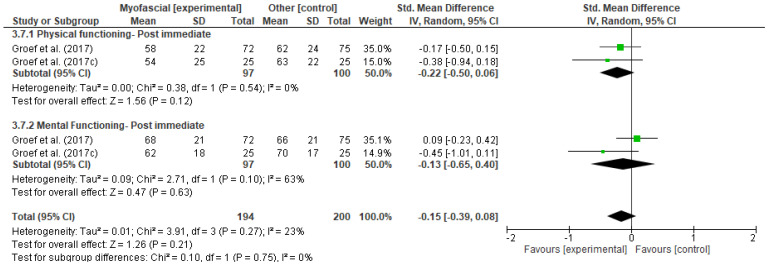
Forest plot of comparison: 3 Myofascial therapy (physical therapy and myofascial therapy) vs. Other intervention (physical therapy and placebo), outcome: 3.7 Quality of life-Post immediate.

**Figure 18 ijerph-18-04420-f018:**
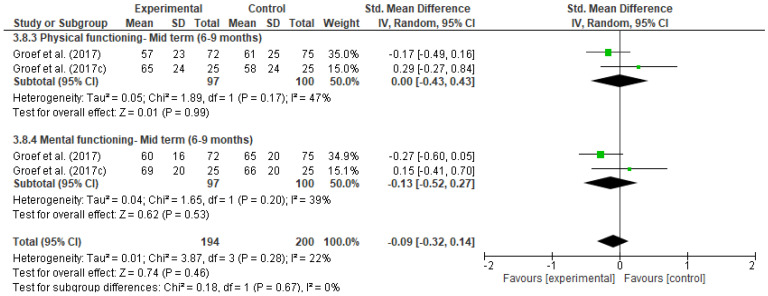
Forest plot of comparison: 3 Myofascial therapy (physical therapy and myofascial therapy) vs. Other intervention (Physical therapy and placebo), outcome: 3.8 Quality of life-Medium term.

**Figure 19 ijerph-18-04420-f019:**
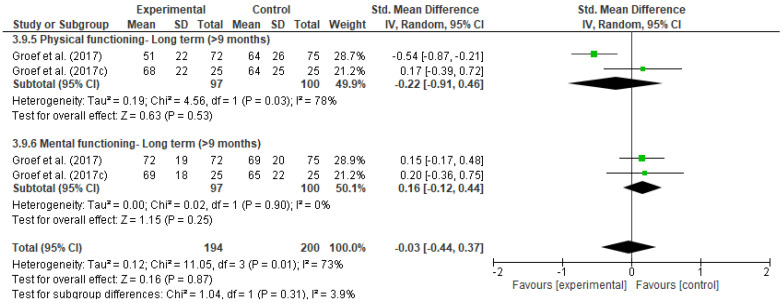
Forest plot of comparison: 3 Myofascial therapy (physical therapy and myofascial therapy) vs. Other intervention (physical therapy and placebo), outcome: 3.9 Quality of life-Long term.

**Table 1 ijerph-18-04420-t001:** Characteristics of the RCTs included in this review.

Author and Year	Sample and Average Age	Diagnosis	Type of Surgical Intervention	Type of Intervention	Sessions and Length of Time	Outcome Measures	Follow-Up	Results
Fernández-Lao et al. (2012) [40] *	N = 20 EG = 10 CG = 10 49 ± 8	Breast cancer (I–IIIA), with moderate/high fatigue (>6 points).	70% of the women had received a lumpectomy and 30% a mastectomy. Adjuvant radiation therapy or chemotherapy.	EG: A protocol of myofascial induction focused on the neck–shoulder area using the Barnes approach. CG: Educational session on healthy lifestyles.	2 sessions 40′/session 3 weeks	PPT: algometer (kg/cm^2^) ATOM (0–45) Function of the sympathetic and immune system: saliva samples	3 weeks	Increased salivary flow (*p* = 0.010) after intervention with myofascial therapy. In addition, positive attitude had a significant increase in IgA (*p* < 0.05) in the experimental group. There were no effects for PPT on the cervical spine or temporalis muscle.
Fernández-Lao et al. (2011) [41] *	N = 20 EG = 10 CG = 10 49 ± 8	Breast cancer (stage I–IIIA) and with moderate/high fatigue (>6 points).	70% of the women had received a lumpectomy and 30% a mastectomy. Adjuvant radiation therapy or chemotherapy.	EG: A protocol of myofascial induction focused on the neck–shoulder area using the Barnes approach. CG: Advice for improving quality of life after breast cancer.	2 sessions 40′/session 2 weeks	HR variability: Holter ATOM (0–45) POMS	2 weeks	Increased HR after myofascial induction (*p* < 0.05) Improvement in perceived fatigue and general mood by POMS after myofascial induction (*p* < 0.05. Improved anxiety, depression and anger in patients with a better attitude towards massage after myofascial therapy (*p* < 0.05).
Groef et al. (2017) [42]	N = 147 EG = 72 CG = 75 EG: 53.9 ± 11.5 CG: 54.7 ± 11.9	Patients with unilateral axillary clearance for primary breast cancer after surgery.	Between 60–70% received a mastectomy and between 30–40% breast conservation. Adjuvant radiation therapy or chemotherapy.	EG: Standard physical therapy program and myofascial induction (in active trigger points of the affected limb and in adhesions of the pectoral, axillary and cervical region, diaphragm and scar). CG: Standard physical therapy intervention and placebo intervention (static hand placement on the upper body and arm).	8 sessions 30′/session 8 weeks	VAS (0–100) DASH (0–100) SF-36 (0–100) PPT: algometer (kg/cm^2^) McGill Pain Questionnaire	8 weeks 9 and 12 months	The PPT in the upper trapezius (*p* = 0.012) was significantly higher at 4 months in the intervention group, and at 4 and 9 months in supraspinatus (*p* = 0.021) and (*p* = 0.040), respectively.
Castro-Martín et al. (2016) [43]	N = 21 EG = 11 CG = 10 25–65	Breast cancer in stage I–IIIA.	The types of surgery were: lumpectomy, quadrantectomy, unilateral mastectomy, mastectomy and lymphadenectomy. All patients had received radiotherapy and chemotherapy.	EG: Myofascial induction in the upper limb area, following the Pilat approach. CG: Simulated pulsed short wave (disconnected)	2 sessions 30′/session 4 weeks	VAS (0–100) Shoulder mobility: Goniometer POMS ATOM Scale (0–45)	4 weeks	The VAS improved after myofascial induction in the affected arm (*p* = 0.031), as well as flexion, abduction and external and internal rotations of the affected arm (*p* < 0.05), and cervical rotation and inclination towards the affected side (*p* < 0.05). After myofascial induction there were also improvements in mood, anxiety, depression, anger, vigor, fatigue and confusion (*p* < 0.05). There were no significant changes on the ATOMS scale.
Serra-añó et al. (2018) [44]	N = 24 EG = 13 CG = 11 EG: 53.15 ± 10.91 CG: 54.36 ± 6.86	Breast cancer with conservative treatment/surgery at least 4 months before, without lymphedema or in stage I.	Conservative/partial surgery with or without stage I lymphedema.	EG: Myofascial induction through four maneuvers. CG: Placebo manual lymphatic drainage	4 sessions 50′/session 4 weeks	VAS (0–100) Shoulder mobility range. DASH (0–100) PHQ-9 (0–100) FACT-B + 4	4 weeks 1 month	Only the participants who received myofascial induction had improved pain intensity, range of motion in flexion, extension, abduction and external rotation of the shoulder (*p* < 0.05), and physical well-being and the general scale of quality of life (*p* < 0.05).
Groef et al. (2017) [45]	N = 50 EG = 23 CG = 25 EG: 55.3 ± 7.5 CG: 53.1 ± 7.5	Unilateral breast cancer with pain (VAS > 4 points) and myofascial dysfunctions in the upper limb area.	Between 60–70% received a mastectomy and between 30–40% breast conservation.	EG: Standard physical therapy program and myofascial induction CG: Standard physical therapy intervention and placebo intervention	20 sessions 60′/session 12 weeks	Shoulder mobility range. Presence of lymphedema Force Dynamics and scapular position: dynamometry. Acromion-table index. Inclinometer. DASH (0–100) SF-36	3 months 12 months	After the intervention, the pain intensity significantly decreased for participants in the experimental group (*p* < 0.046). In the SF-36, mental function improved after myofascial induction (*p* < 0.05).
Groef et al. (2017) [46]	N = 50 EG = 23 CG = 25 EG: 55.36 ± 7.5 CG: 53.1 ± 7.5	Unilateral breast cancer with pain (VAS > 4 points) and myofascial dysfunctions in the upper limb area.	Between 60–70% received a mastectomy and between 30–40% breast conservation.	EG: Standard physical therapy program and myofascial induction CG: Standard physical therapy intervention and placebo intervention	20 sessions 60′/session 12 weeks	VAS (0–100) McGill PPT: algometer (kg/cm^2^) Shoulder functionality: DASH (0–100) SF-36 (0–100)	3 months 6 and 12 months	Increase in the external scale of the scapula in the experimental group (*p* < 0.05) and improvement in physical function related to quality of life (*p* = 0.018).
Massingill et al. (2018) [47]	N = 21 EG = 11 CG = 10 EG/CG: 21 − 55 +	Breast cancer patients who have persistent pain and mobility limitations after breast cancer surgery.	The types of breast cancer surgery included biopsy, lumpectomy, mastectomy or certain types of reconstruction.	EG: Myofascial massage CG: Relaxing massage	Two 30-min sessions a week for 8 weeks	Pain (0–30, with 0 being nothing and 30 being the maximum) Mobility (0–40) Quality of life (0–100)	8 weeks.	The EG experienced more favorable changes in pain than the CG (−10.7 vs. +0.4, *p* < 0.001), mobility (−14.5 vs. −0.8, *p* < 0.001) and overall health (+29.5 vs. −2.5, *p* = 0.002) after 8 weeks

Abbreviations: ATOM, Attitudes Toward Massage Scale; DASH, Disabilities of the Arm, Shoulder, and Hand Questionnaire; HR, Heart Rate; CG, Control Group; EG, Experimental Group; IgA, Immunoglobulin A; POMS, Profile of Mood States; PPT, Pressure Pain Thresholds; SF-36, Health Questionnaire SF-36; VAS, Visual Analog Scale; PHQ-9, Patient Health Questionnaire-9; FACT-B+4, The Functional Assessment of Cancer Therapy for breast cancer patients. * These publications belong to the same study, but evaluate different outcomes.

**Table 2 ijerph-18-04420-t002:** Intervention characteristics of the Myofascial Therapy Group.

Author/Year	N Therapeutic Group	Type	Time Per Session	Number of Sessions	Length of Intervention	Observations
Fernández-Lao et al. (2012) [40] *	10	Myofascial release: the patients received a myofascial release protocol which focused on the neck and shoulder area following the Barnes approach. The protocol included longitudinal strokes, J-strokes, sustained suboccipital pressure, frontal bone decompression and the ear traction technique.	40 min (length adapted to the tissue response of the patient)	2 sessions separated by a 3-week interval	5 weeks	N/A
Fernández-Lao et al. (2011) [41] *	10	Myofascial release: protocol which focused on the neck and shoulder area using the Barnes approach. The protocol included longitudinal strokes, J-strokes, sustained suboccipital pressure, frontal bone decompression and the ear traction technique.	40 min (length adapted to the tissue response of the patient)	2 sessions separated by 2 weeks	4 weeks	80% of the patients underwent surgery at least 12 months before the intervention.
Groef et al. (2017) [42]	72	Standard physical therapy program (shoulder mobilizations, pectoral stretching and relaxation, scar massage, shoulder exercise schemes) + Myofascial therapy consisting of manual myofascial release techniques on (1) active myofascial trigger points in the upper limb area and (2) myofascial adhesions in the pectoral, axillary and cervical regions, diaphragm and scars.	Physical therapy program: 30 min Myofascial release: 30 min	2 sessions a week	8 weeks	The patients were asked to perform exercises twice a day at home. Myofascial interventions were performed from 2 to 4 months after surgery.
Castro-Martín et al. (2016) [43]	21	The patients received a fascial relaxation intervention which focused on the upper limb area, using the Pilat approach.	30 min (length adapted to the tissue response of the patient)	2 sessions separated by a 4-week interval	4 weeks	60% of patients received myofascial intervention in less than 12 months after surgery.
Serra-Añó et al. (2018) [44]	13	The applied treatment was based on the Pilat technique. Four maneuvers were selected for the perinodal and upper thoracic region. They were applied in the following order: sterno-pectoral, global pectoral, pectoral and subscapular.	50 min	1 session a week	4 weeks	The intervention was carried out at least 4 months after surgery.
Groef et al. (2017) [45,46]	25 (2)	Standard physical therapy program (shoulder mobilizations, pectoral stretching and relaxation, scar massage, shoulder exercise schemes) + Myofascial release in (1) active trigger points of the upper limbs and in (2) myofascial adhesions of the pectoralis, axillary and cervical regions, diaphragm and scars.	Physical therapy program: 30 min Myofascial release: 30 min	2 sessions a week (week 1–8) 1 session a week (week 9–12)	12 weeks	The intervention was at least 3 months after radiation therapy and 3 years after surgery.
Massingill et al. (2018) [47]	10	The participants of the intervention received myofascial massage specifically for the chest, thorax and shoulder on the affected side. The massages of the intervention included the following techniques: skin gliding (variable length of time), J-stroke (2–3 min), vertical strokes (2–3 min), strumming (2–3 min), fascial stretching (3–5 min), circular friction (1 to 2 min), deep fascial release (3–5 min), arm pull (60 s on each arm), lateral latissimus dorsi stretch (3 to 5 min) and twisting (3 min).	30 min	2 sessions a week	8 weeks	The intervention began between 3 and 18 months after surgery.

Abbreviations: DN, dry needling; MTrP, myofascial trigger point; NR, not reported; TP, trigger point. * These publications belong to the same study, but evaluate different outcomes.

## Data Availability

All available data can be obtained by contacting the corresponding author.

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
