# Peer review of "Effect of Myofascial Therapy on Pain and Functionality of the Upper Extremities in Breast Cancer Survivors: A Systematic Review and Meta-Analysis"

_ijerph, 2021, doi:10.3390/ijerph18094420_

Round 1
Reviewer 1 Report
Thank you for the opportunity to review this manuscript. It is a very well-organized and thorough review of the effects of myofascial therapy on breast cancer survivors. There are several suggestions for this manuscript:
- Reference is needed for the sentence “The rising incidence of breast cancer in developed countries…” (Page 1, lines 42-44)
- Please check if “hemorrhaging” is a typo (Page 2, lines 56).
- A single sentence cannot generally form a paragraph (Page 2, lines 78-81) (Page 3, lines 110-112) …
- Reference is needed for PICO process (Page 3, page 114)
- Please use the correct heading format for the subheadings under “2.3. Study selection criteria” (e.g., “Types of studies selected:” “Types of participants”)
- Please spell out acronyms, for example, ESA, VAS, DASH, POMS …
- I suggest summarizing/shortening the length of the "Result" section.
- Please consider adding the limitation of this review under the “Discussion” or “Conclusion” sections
Author Response
Journal: IJERPH (ISSN 1660-4601)
Manuscript ID: ijerph-1158001
Title: Effect of Myofascial Therapy on Pain and Functionality of the Upper Extremities in Breast Cancer Survivors: A Systematic Review and Meta-analysis.
We would like to thank the reviewers and the deputy editor for their new comments made. Below please find a detailed list of how we addressed each comment.
Reviewer #1
Thank you for the opportunity to review this manuscript. It is a very well-organized and thorough review of the effects of myofascial therapy on breast cancer survivors. There are several suggestions for this manuscript:
- Reference is needed for the sentence “The rising incidence of breast cancer in developed countries…” (Page 1, lines 39-41)
Comment and response: Thank you for this suggestion. We have inserted a reference (Page 1, line 41, reference 5)
[5] Rojas K, Stuckey A. Breast Cancer Epidemiology and Risk Factors. Clinical Obstetrics and Gynecology. 2016;59(4):651-672.
- Please check if “hemorrhaging” is a typo (Page 2, lines 52).
Comments and response: We are in agreement with the reviewer. We have corrected the typo and we have written “hemorrhage” instead of “hemorrhaging” (Page 2, lines 56).
- A single sentence cannot generally form a paragraph (Page 2, lines 78-81) (Page 3, lines 110-112) …
Comments and response: We are in agreement with the reviewer. We have linked the sentences with the other paragraphs, and we have inserted a new reference (Page 2, lines 72-88, reference 29) (Page 3, lines 101-109): “While improvement in diagnostic processes and in the choices available for medical treatment to reduce possible long-term effects have led to a higher survival rate after breast cancer diagnosis, new challenges have arisen in addressing these effects in healthcare systems [29]. There are several studies which suggest the use of physical therapy to treat side effects of breast cancer following medical treatment. Among the most highly-recommended therapies are mobilization, active exercises or active-assisted exercises, and manual therapy [30-32].”….
- Reference is needed for PICO process (Page 3, page 114)
Comment and response: We are in agreement with the reviewer. We have added reference (Page 3, line 111): “The inclusion and exclusion criteria were defined using the PICO process [39] (Patient, Problem or Population, Intervention, Comparison, Control or Comparator, Outcome(s))”.
[39] da Costa Santos CM, de Mattos Pimenta CA, Nobre MR. The PICO strategy for the research question construction and evidence search. Revista latino-americana de enfermagem, 2007;15(3), 508–511. https://doi.org/10.1590/s0104-11692007000300023
- Please use the correct heading format for the subheadings under “2.3. Study selection criteria” (e.g., “Types of studies selected:” “Types of participants”)
Comment and response: We are grateful for this recommendation. We have numbered the different subtitles within title 2.3. Inclusion and exclusion criteria (Page 3, lines 114-144): “2.3.1. Types of studies; 2.3.2. Types of participants; 2.3.3. Types of interventions; 2.3.4. Types of outcome measures)”
- Please spell out acronyms, for example, ESA, VAS, DASH, POMS …
Comments and response: We are in agreement with the reviewer. We have defined the acronyms from the first time these appear in the text (Page 3, lines 134-144): “For pain, the studies should have used the Visual Analogue Scale (VAS), or a comparable numerical scale, and in the evaluation of functionality, the Disabilities of the Arm, Shoul-der and Hand (DASH), or scale should have been used. As a secondary outcome measure, the following were also taken into consideration: the evaluation of shoulder mobility us-ing goniometry; the mood state of the participants evaluated, for example, with the Profile of Mood States (POMS) scale; and quality of life measured using the Health Questionnaire SF-36. The results were collected in three specific time periods: immediately following treatment, short-term (≤ 3 months), medium-term (between 6-9 months) and long-term (≥ 12 months)”
- I suggest summarizing/shortening the length of the "Result" section.
Comments and response: Thank you for this suggestion. We have modified the results section. On the one hand we have eliminated some comments that are not too important, and that can be seen in the tables. And on the other hand, we have structured the section on the effects of interventions so that it is easier to understand (Page 6, 10-13).
- Please consider adding the limitation of this review under the “Discussion” or “Conclusion”
Comment and response: We are grateful for this recommendation. We have added a new section entitled "Study limitations" after of the discussion (Page 20, lines 486-503): “Our review was limited by the small number published trials to date, the impossibility of blinding participants and physical therapists, the small sample size, and non-homogeneous follow-up of the included studies…”
Reviewer 2 Report
This is review article is a quite good paper. The review is well presented, with fairly solid methodological basis and no particular weak point.
If any, I would suggest to the authors to talk even more about the necessity to carry out new solid studies about the role of Manual Therapy for specific populations. Moreover, I would suggest to highlight better the fact that, to date, there is little scientific consensus on the eventual superiority of some specific Manual Myofascial Treatment methods compared to other specific Manual Myofascial Treatment methods, which is a crucial point in understanding wheter Manual Therapy is a usefull rehabilitation approach for specific conditions/patients.
Author Response
Journal: IJERPH (ISSN 1660-4601)
Manuscript ID: ijerph-1158001
Title: Effect of Myofascial Therapy on Pain and Functionality of the Upper Extremities in Breast Cancer Survivors: A Systematic Review and Meta-analysis.
We would like to thank the reviewers and the deputy editor for their new comments made. Below please find a detailed list of how we addressed each comment.
Reviewer #2
- This is review article is a quite good paper. The review is well presented, with fairly solid methodological basis and no particular weak point.
Comment and response: Thank you for your compliment
- If any, I would suggest to the authors to talk even more about the necessity to carry out new solid studies about the role of Manual Therapy for specific populations. Moreover, I would suggest to highlight better the fact that, to date, there is little scientific consensus on the eventual superiority of some specific Manual Myofascial Treatment methods compared to other specific Manual Myofascial Treatment methods, which is a crucial point in understanding wheter Manual Therapy is a usefull rehabilitation approach for specific conditions/patients.
Comment and response: Thank you for your suggestion. Following this recommendation, we have included the following information on our discussion (page 20, lines 480-484) “Thereby, as shown in this review, there are many different approaches of Myofascial therapy. In order to understand whether myofascial therapy is a useful rehabilitation approach for this population, these differences should also be objective of future randomized controlled trials, to check the superiority of specific methods among others”. We have also emphasized this on our conclusion (page 20, lines 513-515): “future studies are needed to confirm whether myofascial therapy is useful or not to manage with breast cancer survivors’ sequelae”. We hope this highlights the fact that more research is needed to confirm the effects of myofascial therapy.
Reviewer 3 Report
This is a systematic review and meta-analysis of the effect of myofascial therapy on a number of aspects of post-breast cancer treatment. The conclusion is that low to moderate evidence shows no effect of myofascial therapy on pain, function, range of motion or mood.
1.The authors stated purpose is to determine if myofascial therapy is effective in relieving pain, improving functionality, range of motion and mood. As stated, this is a fishing expedition. I would rather see the authors state that their hypothesis was that myofascial release improved these factors, and that the systematic review and meta-analysis with did or did not substantiate the hypothesis.
2. The authors appropriately were PRISM compliant, used the PICO process appropriately in inclusion and exclusion criteria, assessed the risk of bias, and assessed heterogeneity and effect size. There use of Forrest Diagrams was well done.
3. They selected 8 papers, but the meta-analysis was based on 6 papers in their view. However the two papers by Fernandez-Lao (references 39 and 40) are in reality one study. Only the authors reported the effects on mood as a separate study. However, the patient population was the same. In essence, the meta-analysis is based on only 5 studies. The authors should not imply that their systematic review was based on 8 studies, but should clearly state that the meta-analysis is based on 6 publications and represents 5 studies.
4. The results that are given in narrative form in the Results section are listed in long sentences with each factor separated by a semi-colon. This is in fact hard to read. The authors would do better listing each as a separate sentence, perhaps bulleted, or just list the results in a table and refer to the table in the results section.
5. The means of myofascial release vary, as the authors indicate. However, all are manual. That should be made clear.
6. The discussion is perhaps overlong for a paper in which the conclusion is that the level of evidence is low to moderate for the most part and that current evidence does not support the use of myofascial release and the studies on which this is based are of poor quality and varied.
7. Likewise, the opening sentences in the introduction about the epidemiology of cancer are not really needed for this kind of paper. The authors can go right to the topic of symptom relief after breast cancer treatment and discuss the reason that they looked at myofascial therapy.
8. Does considering references 39 and 40 as one study (although looking at different factors) change the statistics at all? For one thing, it reduces the total number of subjects studied in all of the selected studies by 20 subjects.
Author Response
Journal: IJERPH (ISSN 1660-4601)
Manuscript ID: ijerph-1158001
Title: Effect of Myofascial Therapy on Pain and Functionality of the Upper Extremities in Breast Cancer Survivors: A Systematic Review and Meta-analysis.
We would like to thank the reviewers and the deputy editor for their new comments made. Below please find a detailed list of how we addressed each comment.
Reviewer #3
This is a systematic review and meta-analysis of the effect of myofascial therapy on a number of aspects of post-breast cancer treatment. The conclusion is that low to moderate evidence shows no effect of myofascial therapy on pain, function, range of motion or mood.
- The authors stated purpose is to determine if myofascial therapy is effective in relieving pain, improving functionality, range of motion and mood. As stated, this is a fishing expedition. I would rather see the authors state that their hypothesis was that myofascial release improved these factors, and that the systematic review and meta-analysis with did or did not substantiate the hypothesis.
Comment and response: Following your useful suggestion, we have added to the last paragraph of the introduction the following sentence (page 2, lines 91-90) “we hypothesize that myofascial release is an adequate approach to improve these factors”. Moreover, we have indicated that “future studies are needed to confirm whether myofascial therapy is useful or not to manage breast cancer survivors’ sequelae” in our conclusion (page 20, lines 514-516). We appreciate this suggestion and hope this sentence enhances the objective of our manuscript.
- The authors appropriately were PRISM compliant, used the PICO process appropriately in inclusion and exclusion criteria, assessed the risk of bias, and assessed heterogeneity and effect size. There use of Forrest Diagrams was well done.
Comment and response: We thank the reviewer for these positive comments.
- They selected 8 papers, but the meta-analysis was based on 6 papers in their view. However the two papers by Fernandez-Lao (references 39 and 40) are in reality one study. Only the authors reported the effects on mood as a separate study. However, the patient population was the same. In essence, the meta-analysis is based on only 5 studies. The authors should not imply that their systematic review was based on 8 studies, but should clearly state that the meta-analysis is based on 6 publications and represents 5 studies.
Comment and response: Thank you for your appreciation. We have revised this and change it when necessary through the paper to not to confuse the reader. We have also added and explanation for this below tables 1 and 2.
- The results that are given in narrative form in the Results section are listed in long sentences with each factor separated by a semi-colon. This is in fact hard to read. The authors would do better listing each as a separate sentence, perhaps bulleted, or just list the results in a table and refer to the table in the results section.
Comments and response: Thank you for this suggestion. We have modified the results section, we have structured the section on the effects of interventions so that it is easier to understand (Page 11-13, lines 290-318, lines 322-368): “3.8. Effects of the interventions; 3.8.1. Myofascial therapy vs placebo treatment or other intervention at post-immediate. Analysis 4 to 7 show the estimated primary effect size (4 to 8 weeks post-treatment) of the intervention with myofascial therapy alone or combined, compared to placebo treatment or other intervention with physical therapy for the outcomes of pain intensity, functionality, and range of motion in flexion and abduction.
- Pain Intensity. This sub-analysis included 4 trials [42-45] with 242 participants. No significant differences were observed between the effects of myofascial therapy alone or in combination with a standard physical therapy program and placebo intervention or a standard physical therapy program (Analysis 4. SMD -0.15, 95% CI -0.48 to 0.19, I² = 28%)…” In each of the comparisons we have written separately the results for each variable.
- The means of myofascial release vary, as the authors indicate. However, all are manual. That should be made clear.
Comments and response: Thank you for your appreciation. On the subsection 3.4, characteristics of the interventions (page 6, lines 224-225), we have indicated, to make it clearer, the following sentence: “The types of myofascial intervention used, although all of them were manual interventions, varied among the studies”. We hope now with this sentence this information is clearer.
- The discussion is perhaps overlong for a paper in which the conclusion is that the level of evidence is low to moderate for the most part and that current evidence does not support the use of myofascial release and the studies on which this is based are of poor quality and varied.
Comments and response: Thank you for this suggestion. We have shortened the section "4.1. Summary of the evidence", highlighting only the most relevant findings (page 19, lines 420-446): “…There were greater general effects in favor of myofascial therapy than other controls/interventions, but the subgroup analyses revealed inconsistent, insignificant results. Only the range of motion in abduction showed statistical significance in favor of the experimental group. Here, myofascial therapy is at least equivalent to other forms of intervention.
Several factors should be considered when interpreting our findings for clinical recommendation and implementation. Although there was no statistical significance, all of the studies individually reported a clinically relevant improvement in favor of the experimental group with myofascial intervention. The effectiveness of myofascial therapy is unclear because the subgroup analyses were hampered by the small number of studies that included the same outcome measures and follow-up beyond immediately following the intervention. Very few studies reported comparable estimates regarding secondary outcome measures to calculate the effect of using myofascial therapy”.
In addition, we have added a section on study limitations (Page 20, lines 487-503): “Our review was limited by the small number published trials to date, the impossibility of blinding participants and physical therapists, the small sample size, and non-homogeneous follow-up of the included studies…”
We appreciate your opinion, and we agree that the evidence is very varied and of low methodological quality, this is one of the main limitations that we have found when performing this review, to date there are few RCTs evaluating myofascial therapy in patients with cancer beyond the post-immediate follow up.
- Likewise, the opening sentences in the introduction about the epidemiology of cancer are not really needed for this kind of paper. The authors can go right to the topic of symptom relief after breast cancer treatment and discuss the reason that they looked at myofascial therapy.
Comments and response: Thank you for your comment. We have reduced the amount of information of the first paragraph related to incidence to focus on breast cancer incidence. Thereby we have simplified the information about the medical treatment. We believe this information is of interest for the reader to better understand the damage suffered by the tissues and therefore the need of a rehabilitation treatment after it. At the end of the introduction we have included a sentence to justify why we looked at myofascial therapy for this review (pages 2, lines 86-88).
- Does considering references 39 and 40 as one study (although looking at different factors) change the statistics at all? For one thing, it reduces the total number of subjects studied in all of the selected studies by 20 subjects
Comments and response: We are really thankful for this appreciation. Consequently, we have modified this information on the subsection “characteristics of studies included” as follows (page 5, lines 195-197): “Eight RCTs were included, with a total of 333 participants [40-47]. The study developed by Fernández-Lao et al., divided into 2 publications [40,41] contained the smallest sample-size, with 20 participants”. We have done similar on the discussion (page 19, lines 425-428), to not to confuse the reader.
Regarding your interesting comment about the change of the statistics for this change on the sample size, as both studies are not included together in any of the analysis performed, this change does not modify the results obtained on our meta-analysis.